# IMPROVING END-TO-END OBJECT TRACKING USING RELATIONAL REASONING

## ABSTRACT

Relational reasoning—the ability to model interactions and relations between objects—is valuable for robust multi-object tracking and pivotal for trajectory prediction. In this paper we propose MOHART, a class-agnostic, end-to-end multi-object tracking and trajectory prediction algorithm, which explicitly accounts for permutation invariance in its relational reasoning. We explore a number of permutation invariant architectures and show that multi-headed self-attention outperforms the provided baselines and better accounts for complex physical interactions in a challenging toy experiment. We show on three real-world tracking datasets that adding relational reasoning capabilities in this way increases the tracking and trajectory prediction performance, particularly in the presence of ego-motion, occlusions, crowded scenes, and faulty sensor inputs. To the best of our knowledge, MOHART is the first fully end-to-end multi-object tracking from vision approach applied to real-world data reported in the literature.

## 1 INTRODUCTION

Real-world environments can be rich and contain countless types of interacting objects. Intelligent autonomous agents need to understand both the objects and interactions between them if they are to operate in those environments. This motivates the need for *class-agnostic* algorithms for tracking multiple objects—a capability that is not supported by the popular tracking-by-detection paradigm. In tracking-by-detection, objects are detected in each frame independently, e. g., by a pre-trained deep convolutional neural network (CNN) such as YOLO (Redmon et al. (2016)), and then linked across frames. Algorithms from this family

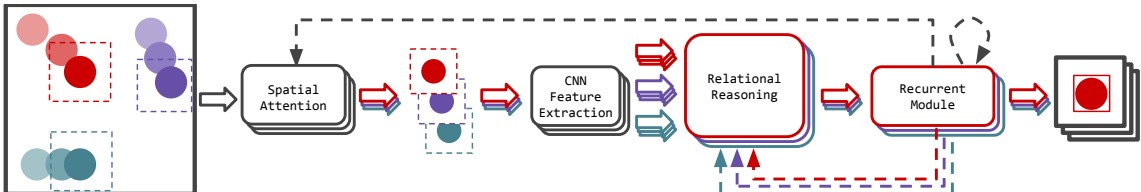

FIGURE 1: Multi-object hierarchical attentive recurrent tracking (MOHART). A glimpse is extracted for each object using a (fully differentiable) spatial attention mechanism. These glimpses are further processed with a CNN and fed into a relational reasoning module. A recurrent module which iterates over time steps allows for capturing of complex motion patterns. It also outputs spatial attention parameters and a feature vector per object for the relational reasoning module. Dashed lines indicate temporal connections (from time step $t$ to $t + 1$). The entire pipeline operates in parallel for the different objects, only the relational reasoning module allows for exchange of information between tracking states of each object. MOHART is an extension of HART (a single-object tracker), which features the same pipeline without the relational reasoning module.

can achieve high accuracy, provided sufficient labelled data to train the object detector, and given that all encountered objects can be associated with known classes, but fail when faced with objects from previously unseen categories.

Hierarchical attentive recurrent tracking (HART) is a recently-proposed, alternative method for single-object tracking (SOT), which can track arbitrary objects indicated by the user (Kosiorek et al. (2017)). This is done by providing an initial bounding-box, which may be placed over any part of the image, regardless of whether it contains an object or what class the object is. HART efficiently processes just the relevant part of an image using spatial attention; it also integrates object detection, feature extraction, and motion modelling into one network, which is trained fully end-to-end. Contrary to tracking-by-detection, where only one video frame is typically processed at any given time to generate bounding box proposals, end-to-end learning in HART allows for discovering complex visual and spatio-temporal patterns in videos, which is conducive to inferring what an object is and how it moves.

In the original formulation, HART is limited to the single-object modality—as are other existing end-to-end trackers (Kahou et al. (2017); Rasouli Danesh et al. (2019); Gordon et al. (2018)). In this work, we present MOHART, a class-agnostic tracker with complex relational reasoning capabilities provided by a multi-headed self-attention module (Vaswani et al. (2017); Lee et al. (2019)). MOHART infers the latent state of every tracked object in parallel, and uses self-attention to inform per-object states about other tracked objects. This helps to avoid performance loss under self-occlusions of tracked objects or strong camera motion. Moreover, since the model is trained end-to-end, it is able to learn how to manage faulty or missing sensor inputs. See fig. 1 for a high-level illustration of MOHART.

In order to track objects, MOHART estimates their states, which can be naturally used to predict future trajectories over short temporal horizons, which is especially useful for planning in the context of autonomous agents. MOHART can be trained simultaneously for object tracking and trajectory prediction at the same time, thereby increasing statistical efficiency of learning. In contrast to prior art, where trajectory prediction and object tracking are usually addressed as separate problems with unrelated solutions, our work show trajectory prediction and object tracking are best addressed jointly.

Section 2 describes prior art in tracking-by-detection, end-to-end tracking and pedestrian trajectory prediction. In Section 3, we describe our approach, which uses a permutation-invariant self-attention module to enable tracking multiple objects end-to-end with relational reasoning. Section 4 contrasts our approach with multi-object trackers which do not explicitly enforce permutation invariance but have the capacity to learn it, simpler permutation-invariant architectures, as well as multiple single-object trackers running in parallel. We show that multi-headed self-attention significantly outperforms other approaches. Finally, in Section 5, we apply MOHART to real world datasets and show that permutation-invariant relational reasoning leads to consistent performance improvement compared to HART both in tracking and trajectory prediction.

## 2 RELATED WORK

**Tracking-by-Detection**  Vision-based tracking approaches typically follow a tracking-by-detection paradigm: objects are first detected in each frame independently, and then a tracking algorithm links the detections from different frames to propose a coherent trajectory (Zhang et al. (2008); Milan et al. (2014); Bae and Yoon (2017); Keuper et al. (2018)). Motion models and appearance are often used to improve the association between detected bounding-boxes in a postprocessing step. Tracking-by-detection algorithms currently provide the state-of-the-art in multi-object tracking on common benchmark suites, and we fully acknowledge that MOHART is not competitive at this stage in scenarios where high-quality detections are available for each frame. MOHART can in principle be equipped with the ability to use bounding boxes provided by an object detector, but this is beyond the scope of this project.

**End-to-End Tracking**   A newly established and much less explored stream of work approaches tracking in an end-to-end fashion. A key difficulty here is that extracting an image crop (according to bounding-boxes provided by a detector), is non-differentiable and results in high-variance gradient estimators. Kahou et al. (2017) propose an end-to-end tracker with soft spatial-attention using a 2D grid of Gaussians instead of a hard bounding-box. HART draws inspiration from this idea, employs an additional attention mechanism, and shows promising performance on the real-world KITTI dataset (Kosiorek et al. (2017)). HART forms the foundation of this work. It has also been extended to incorporate depth information from RGBD cameras (Rasouli Danesh et al. (2019)). Gordon et al. (2018) propose an approach in which the crop corresponds to the scaled up previous bounding-box. This simplifies the approach, but does not allow the model to learn where to look— i. e., no gradient is backpropagated through crop coordinates. To the best of our knowledge, there are no successful implementations of any such end-to-end approaches for multi-object tracking beyond SQAIR (Kosiorek et al. (2018)), which works only on datasets with static backgrounds. On real-world data, the only end-to-end approaches correspond to applying multiple single-object trackers in parallel—a method which does not leverage the potential of scene context or inter-object interactions.

**Pedestrian trajectory prediction**   Predicting pedestrian trajectories has a long history in computer vision and robotics. Initial research modelled social forces using hand-crafted features (Lerner et al. (2007); Pellegrini et al. (2009); Trautman and Krause (2010); Yamaguchi et al. (2011)) or MDP-based motion transition models (Rudenko et al. (2018)), while more recent approaches learn from context information, e. g., positions of other pedestrians or landmarks in the environment. Social-LSTM (Alahi et al. (2016)) employs a long short-term memory (LSTM) to predict pedestrian trajectories and uses max-pooling to model global social context. Attention mechanisms have been employed to query the most relevant information, such as neighbouring pedestrians, in a learnable fashion (Su et al. (2016); Fernando et al. (2018); Sadeghian et al. (2019)). Apart from relational learning, context (Varshneya and Srinivasaraghavan (2017)), periodical time information (Sun et al. (2018)), and constant motion priors (Schöller et al. (2019)) have proven effective in predicting long-term trajectories.

Our work stands apart from this prior art by not relying on ground truth tracklets. It addresses the more challenging task of working directly with visual input, performing tracking, modelling interactions, and, depending on the application scenario, simultaneously predicting future motions. As such, it can also be compared to Visual Interaction Networks (VIN) (Watters et al. (2017)), which use a CNN to encode three consecutive frames into state vectors—one per object—and feed these into a recurrent neural network (RNN), which has an Interaction Network (Battaglia et al. (2016)) at its core. More recently, Relational Neural Expectation Maximization (R-NEM) has been proposed as an unsupervised approach which combines scene segmentation and relational reasoning (van Steenkiste et al. (2018)). Both VINs and R-NEM make accurate predictions in physical scenarios, but, to the best of our knowledge, have not been applied to real world data.

## 3   RECURRENT MULTI-OBJECT TRACKING WITH SELF-ATTENTION

This section describes the model architecture in fig. 1. We start by describing the hierarchical attentive recurrent tracking (HART) algorithm (Kosiorek et al. (2017)), and then follow with an extension of HART to tracking multiple objects, where multiple instances of HART communicate with each other using multi-headed attention to facilitate relational reasoning. We also explain how this method can be extended to trajectory prediction instead of just tracking.

### 3.1   HIERARCHICAL ATTENTIVE RECURRENT TRACKING (HART)

HART is an attention-based recurrent algorithm, which can efficiently track single objects in a video. It uses a spatial attention mechanism to extract a *glimpse* $\mathbf{g}_t$, which corresponds to a small crop of the image $\mathbf{x}_t$ at time-step $t$, containing the object of interest. This allows it to dispense with the processing of the whole image and can significantly decrease the amount of computation required. HART uses a CNN to convert the

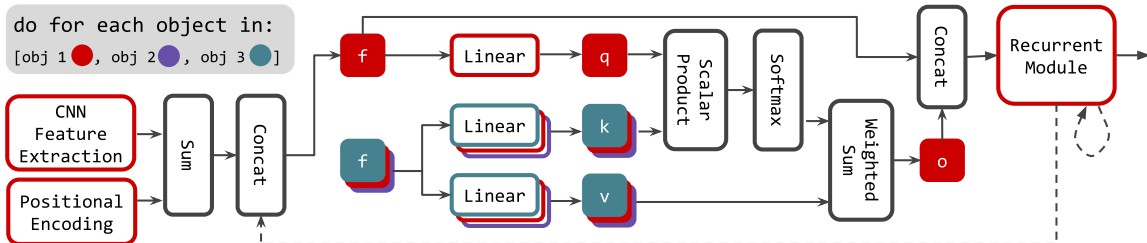

FIGURE 2: The relational reasoning module in MOHART based on multi-headed self-attention. Here, we show the computation of the interaction of the red object with all other objects. Object representations $f_{t,m}$ are computed using visual features, positional encoding and the hidden state from the recurrent module. These are linearly projected onto keys (k), queries (q), and values (v) to compute a weighted sum of interactions between objects, yielding an interaction vector $o_{t,m}$. Subscripts $t, m$ are dropped from all variables for clarity of presentation, so is the splitting into multiple heads.

glimpse $\mathbf{g}_t$ into features $\mathbf{f}_t$, which then update the hidden state $\mathbf{h}_t$ of a LSTM core. The hidden state is used to estimate the current bounding-box $\mathbf{b}_t$, spatial attention parameters for the next time-step $\mathbf{a}_{t+1}$, as well as object appearance. Importantly, the recurrent core can learn to predict complicated motion conditioned on the past history of the tracked object, which leads to relatively small attention glimpses—contrary to CNN-based approaches (Held et al. (2016); Valmadre et al. (2017)), HART does not need to analyse large regions-of-interest to search for tracked objects. In the original paper, HART processes the glimpse with an additional ventral and dorsal stream on top of the feature extractor. Early experiments have shown that this does not improve performance on the MOTChallenge dataset, presumably due to the oftentimes small objects and overall small amount of training data. Further details are provided in Appendix B.

The algorithm is initialised with a bounding-box[1] $\mathbf{b}_1$ for the first time-step, and operates on a sequence of raw images $\mathbf{x}_{1:T}$. For time-steps $t \geq 2$, it recursively outputs bounding-box estimates for the current time-step and predicted attention parameters for the next time-step. The performance is measured as intersection-over-union (IoU) averaged over all time steps in which an object is present, excluding the first time step.

Although HART can track arbitrary objects, it is limited to tracking one object at a time. While it can be deployed on several objects in parallel, different HART instances have no means of communication. This results in performance loss, as it is more difficult to identify occlusions, ego-motion and object interactions. Below, we propose an extension of HART which remedies these shortcomings.

### 3.2 MULTI-OBJECT HIERARCHICAL ATTENTIVE RECURRENT TRACKING (MOHART)

Multi-object support in HART requires the following modifications. Firstly, in order to handle a dynamically changing number of objects, we apply HART to multiple objects in parallel, where all parameters between HART instances are shared. We refer to each HART instance as a *tracker*. Secondly, we introduce a presence variable $p_{t,m}$ for object $m$. It is used to mark whether an object should interact with other objects, as well as to mask the loss function (described in (Kosiorek et al. (2017))) for the given object when it is not present. In this setup, parallel trackers cannot exchange information and are conceptually still single-object trackers, which we use as a baseline, referred to as HART (despite it being an extension of the original algorithm). Finally, to enable communication between trackers, we augment HART with an additional step between feature extraction and the LSTM.

---

[1]We can use either a ground-truth bounding-box or one provided by an external detector; the only requirement is that it contains the object of interest.

For each object, a glimpse is extracted and processed by a CNN (see fig. 1). Furthermore, spatial attention parameters are linearly projected on a vector of the same size and added to this representation, acting as a positional encoding. This is then concatenated with the hidden state of the recurrent module of the respective object (see fig. 2). Let $\mathbf{f}_{t,m}$ denote the resulting feature vector corresponding to the m[th] object, and let $\mathbf{f}_{t,1:M}$ be the set of such features for all objects. Since different objects can interact with each other, it is necessary to use a method that can inform each object about the effects of their interactions with other objects. Moreover, since features extracted from different objects comprise a set, this method should be permutation-equivariant, i. e., the results should not depend on the order in which object features are processed. Therefore, we use the multi-head self-attention block (SAB, Lee et al. (2019)), which is able to account for higher-order interactions between set elements when computing their representations. Intuitively, in our case, SAB allows any of the trackers to query other trackers about attributes of their respective objects, e. g., distance between objects, their direction of movement, or their relation to the camera. This is implemented as follows,

$$Q = W_q \mathbf{f}_{1:M} + b_q \,, \qquad K = W_k \mathbf{f}_{1:M} + b_k \,, \qquad V = W_v \mathbf{f}_{1:M} + b_v \,, \tag{1}$$

$$O_i = \text{softmax}\left(Q_i K_i^T\right) V_i \,, \qquad i = 1, \ldots, H \,, \tag{2}$$

$$o_{1:M} = O = \text{concat}(O_i, \ldots, O_H) \,, \tag{3}$$

where $o_m$ is the output of the relational reasoning module for object $m$. Time-step subscripts are dropped to decrease clutter. In Eq. 1, each of the extracted features $\mathbf{f}_{t,m}$ is linearly projected into a triplet of key $\mathbf{k}_{t,m}$, query $\mathbf{q}_{t,m}$ and value $\mathbf{v}_{t,m}$ vectors. Together, they comprise $K, Q$ and $V$ matrices with $M$ rows and $d_q, d_k, d_k$ columns, respectively. $K, Q$ and $V$ are then split up into multiple heads $H \in \mathbb{N}_+$, which allows to query different attributes by comparing and aggregating different projection of features. Multiplying $Q_i K_i^T$ in Eq. 2 allows to compare every query vector $\mathbf{q}_{t,m,i}$ to all key vectors $\mathbf{k}_{t,1:M,i}$, where the value of the corresponding dot-products represents the degree of similarity. Similarities are then normalised via a $\text{softmax}$ operation and used to aggregate values $V$. Finally, outputs of different attention heads are concatenated in Eq. 3. SAB produces $M$ output vectors, one for each input, which are then concatenated with corresponding inputs and fed into separate LSTMs for further processing, as in HART—see fig. 1.

MOHART is trained fully end-to-end, contrary to other tracking approaches. It maintains a hidden state, which can contain information about the object's motion. One benefit is that in order to predict future trajectories, one can simply feed black frames into the model. Our experiments show that the model learns to fall back on the motion model captured by the LSTM in this case.

### 3.3 MULTI-OBJECT BASELINES

**Multilayer perceptron (MLP)** In this version, the representations of all objects are concatenated and fed into a fully connected layer followed by ELU activations. The output is then again concatenated to the unaltered feature vector of each object. This concatenated version is then fed to the recurrent module of HART. This way of exchanging information allows for universal function approximation (in the limit of infinite layer sizes) but does not impose permutation invariance.

**DeepSets** Here, the learned representations of the different objects are summed up instead of concatenated and then divided by total number of objects. This is closely related to DeepSets (Zaheer et al. (2017)) and allows for universal function approximation of all permutation invariant functions (Wagstaff et al. (2019)).

**Max-Pooling** Similar to DeepSets, but using max-pooling as the permutation invariant operation. This way of exchanging information is used, e.g., by Alahi et al. (2016) who predict future pedestrian trajectories from ground truth tracklets in coordinate space.

### 4 VALIDATION ON SIMULATED DATA

We test and compare the relational reasoning capabilities of the proposed algorithms on a toy domain. The domain is a 2D squared box which contains circular objects with approximated elastic collisions (energy

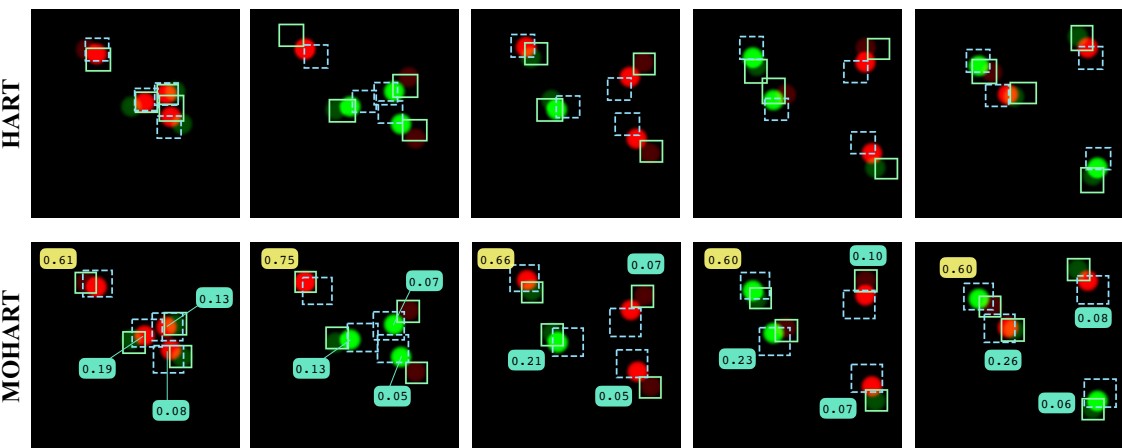

FIGURE 3: A scenario constructed to be impossible to solve without relational reasoning. Circles of the same colour repel each other, circles of different colour attract each other. Crucially, each circle is randomly assigned its identity in each time step. Hence, the algorithm can not infer the forces exerted on one object without knowledge of the state of the other objects in the current time step. The forces in this scenario scale with $1/\sqrt{r}$ and the algorithm was trained to predict one time step into the future. HART (top) is indeed unable to predict the future location of the objects accurately. The achieved average IoU is $47\%$, which is only slightly higher than predicting the objects to have the same position in the next time step as in the current one ($34\%$). Using the relational reasoning module, MOHART (bottom) is able to make meaningful predictions ($76\%$ IoU). The numbers in the bottom row indicate the self-attention weights from the perspective of the top left tracker (yellow number box). Interestingly, the attention scores have a strong correlation with the interaction strength (which scales with distance) without receiving supervision.

and momentum conservation) between objects and with walls. To investigate how the model understands motion patterns and interactions between objects, we train it to predict future object locations in contrast to traditional tracking.

In the first experiment, each circle exerts repulsive forces on each other, where the force scales with $1/r$, $r$ being the distance between them. Predicting the future location just using the previous motion of one object (i.e. without relational reasoning) accurately is therefore challenging. We show that HART as an end-to-end single-object tracker is able to capture complex motion patterns and leverage these to make accurate predictions (see Appendix C). This indicates that HART is able to draw conclusions about the (deterministic, but not static) force field.

In the second experiment, we introduce randomness, rendering the scenario not solvable for a single object tracker as it requires knowledge about the state of the other objects and relational reasoning (see fig. 3). In each time step, we assign a colour-coded identity to the objects. Objects of the same identity repel each other, object of different identities attract each other (the objects can be thought of as electrons and protons). The qualitative results in fig. 3 show that MOHART, using self-attention for relational reasoning, is able to capture these interactions with high accuracy. Figure 4 (left) shows a quantitative comparison of augmenting HART with different relational reasoning modules when identities are re-assigned in every timestep (randomness $= 1.0$). Exchanging information between trackers of different objects in the latent space with an MLP leads to slightly worse performance than the HART baseline, while simple max-pooling performs significantly better ($\Delta\mathrm{IoU} \sim 17\%$). This can be explained through the permutation invariance of the problem: the list of latent representation of the different objects has no meaningful order and the output of the model should therefore be invariant to the ordering of the objects. The MLP is in itself not permutation

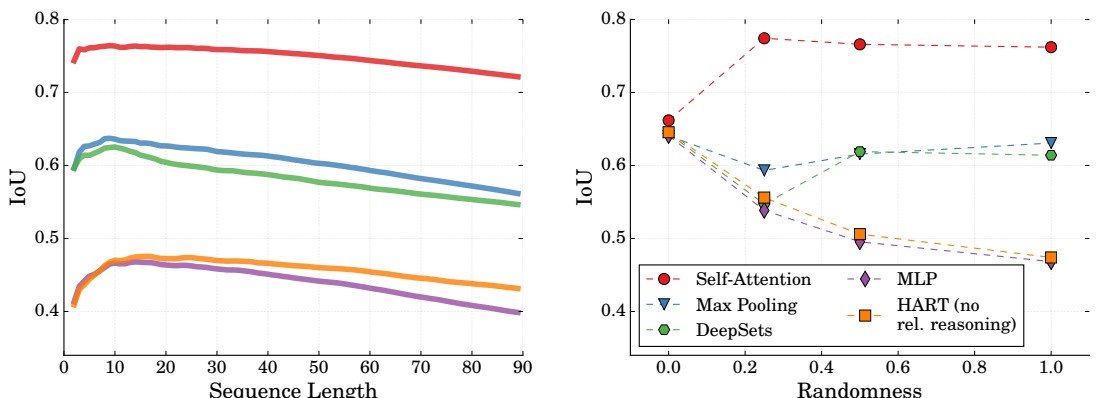

FIGURE 4: Left: average IoU over sequence length for different implementations of relational reasoning on the toy domain shown in fig. 3 (randomness = 1.0). Right: performance depending on how often agents are re-assigned identities randomly (sequence length 15). The higher the randomness, the less static the force field is and the more vital relational reasoning is. For randomness = 0.0, identities still have to be reassigned in some cases in order to prevent deadlocks, this leads to a performance loss for all models, which explains lower performance of self-attention for randomness = 0.0.

invariant and therefore prone to overfit to the (meaningless) order of the objects in the training data. Max-pooling, however, is permutation invariant and can in theory, despite its simplicity, be used to approximate any permutation invariant function given a sufficiently large latent space (Wagstaff et al. (2019)). Max-pooling is often used to exchange information between different tracklets, e.g., in the trajectory prediction domain (Alahi et al. (2016); Gupta et al. (2018)). However, self-attention, allowing for learned querying and encoding of information, solves the relational reasoning task significantly more accurately. In fig. 4 (right), the frequency with which object identities are reassigned randomly is varied. The results show that, in a deterministic environment, tracking does not necessarily profit from relational reasoning - even in the presence of long-range interactions. The less random, the more static the force field is and a static force field can be inferred from a small number of observations (see fig. 6). This does not mean that all stochastic environments profit from relational reasoning. What these experiments indicate is that tracking can not be expected to profit from relational reasoning by default in any environment, but instead in environments which feature (potentially non-deterministic) dynamics and predictable interactions.

## 5 RELATIONAL REASONING IN REAL-WORLD TRACKING

Having established that MOHART is capable of performing complex relational reasoning, we now test the algorithm on three real world datasets and analyse the effects of relational reasoning on performance depending on dataset and task. We find consistent improvements of MOHART compared to HART throughout. Relational reasoning yields particularly high gains for scenes with ego-motion, crowded scenes, and simulated faulty sensor inputs.

### 5.1 EXPERIMENTAL DETAILS

We investigate three qualitatively different datasets: the MOTChallenge dataset (Milan et al. (2016)), the UA-DETRAC dataset (Wen et al. (2015)), and the Stanford Drone dataset (Robicquet et al. (2016)). To increase scene dynamics and make the tracking/prediction problems more challenging, we sub-sample some of the high framerate scenes with a stride of two, resulting in scenes with 7-15 frames per second. Training and architecture details are given in Appendices A and B. We conduct experiments in three different modes:

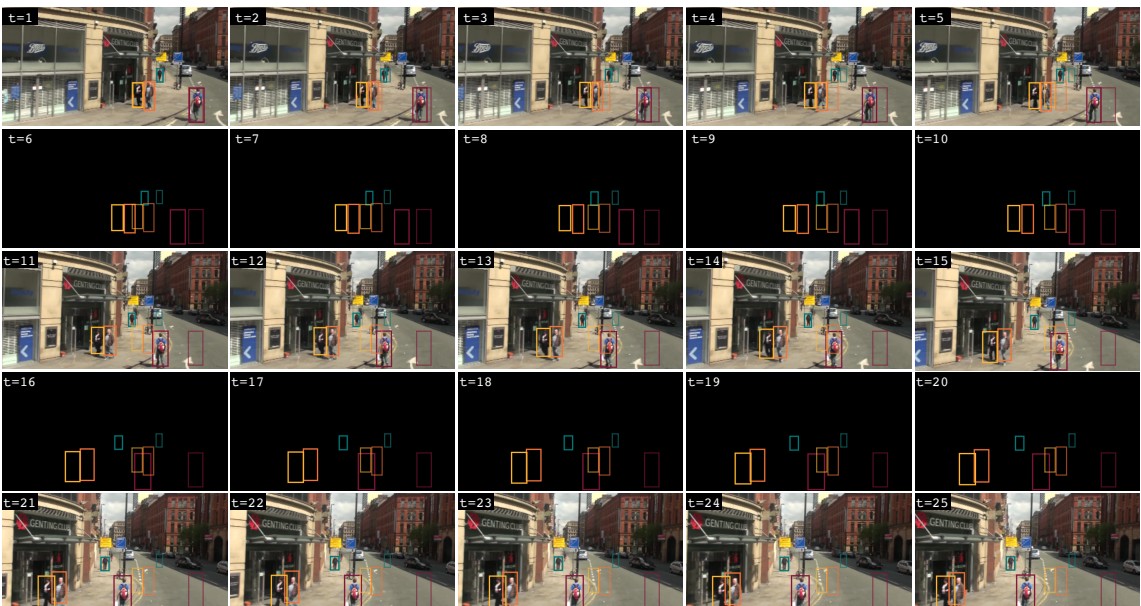

FIGURE 5: Camera blackout experiment on a street scene from the MOTChallenge dataset with strong ego-motion. Solid boxes are MOHART predictions (for $t \geq 2$), faded bounding boxes indicate object locations in the first frame. As the model is trained end-to-end, MOHART learns to fall back onto its internal motion model if no new observations are available (black frames). As soon as new observations come in, the model 'snaps' back onto the tracked objects.

**Tracking.** The model is initialised with the ground truth bounding boxes for a set of objects in the first frame. It then consecutively sees the following frames and predicts the bounding boxes. The sequence length is 30 time steps and the performance is measured as intersection over union (IoU) averaged over the entire sequence excluding the first frame. This algorithm is either applied to the entire dataset or subsets of it to study the influence of certain properties of the data.

**Camera Blackout.** This simulates unsteady or faulty sensor inputs. The setup is the same as in *Tracking*, but sub-sequences of the input are replaced with black images. The algorithm is expected to recognise that no new information is available and that it should resort to its internal motion model.

**Prediction.** Testing MOHART's ability to capture motion patterns, only the first two frames are shown to the model followed by three black frames. IoU is measured seperately for each time step.

## 5.2 RESULTS AND ANALYSIS

On the MOTChallenge dataset, HART achieves $66.6\%$ intersection over union (see Table 1), which in itself is impressive given the small amount of training data of only 5225 training frames and no pre-training. MOHART achieves $68.5\%$ (both numbers are averaged over 5 runs, independent samples $t$-test resulted in $p < 0.0001$). The performance gain increases when only considering ego-motion data. This is readily explained: movements of objects in the image space due to ego-motion are correlated and can therefore be better understood when combining information from movements of multiple objects, i.e. performing relational reasoning. In another ablation, we filtered for only crowded scenes by requesting five objects to be present for, on average, 90% of the frames in a sub-sequence. For the MOT-Challenge dataset, this only

TABLE 1: Tracking performance on the MOTChallenge dataset measured in IoU.

| | Entire Dataset | Only Ego-Motion | No Ego-Motion | Crowded Scenes | Camera Blackout |
|---|---|---|---|---|---|
| **MOHART** | **68.5%** | **66.9%** | **64.7%** | **69.1%** | **63.6%** |
| HART | 66.6% | 64.0% | 62.9% | 66.9% | 60.6% |
| Δ | 1.9% | 2.9% | 1.8% | 2.2% | 3.0% |

TABLE 2: UA-DETRAC Dataset

| | All | Crowded Scenes | Camera Blackout |
|---|---|---|---|
| **MOHART** | 68.1% | **69.5%** | **64.2%** |
| HART | **68.4%** | 68.6% | 53.8% |
| Δ | -0.3% | 0.9% | 0.4% |

TABLE 3: Stanford Drone Dataset

| | All | Camera Blackout | CamBlack Bikes |
|---|---|---|---|
| **MOHART** | **57.3%** | **53.3%** | **53.3%** |
| HART | 56.1% | 52.6% | 50.7% |
| Δ | 1.2% | 0.7% | 2.6% |

leads to a minor increase of the performance gain of MOHART indicating that the dataset exhibits a sufficient density of objects to learn interactions. The biggest benefit from relational reasoning can be observed in the *camera blackout* experiments (setup explained in Section 5.1). Both HART and MOHART learn to rely on their internal motion models when confronted with black frames and propagate the bounding boxes according to the previous movement of the objects. It is unsurprising that this scenario profits particularly from relational reasoning. Qualitative tracking and *camera blackout* results are shown in fig. 5 and Appendix E.

Tracking performance on the UA-DETRAC dataset only profits from relational reasoning when filtering for crowded scenes (see Table 2). The fact that the performance of MOHART is slightly worse on the vanilla dataset ($\Delta = -0.3\%$) can be explained with more overfitting. As there is no exchange between trackers for each object, each object constitutes an independent training sample.

The Stanford drone dataset (see Table 3) is different to the other two—it is filmed from a birds-eye view. The scenes are more crowded and each object covers a small number of pixels, rendering it a difficult problem for tracking. The dataset was designed for trajectory prediction—a setup where an algorithm is typically provided with ground-truth tracklets in coordinate space and potentially an image as context information. The task is then to extrapolate these tracklets into the future. The tracking performance profits from relational reasoning more than on the UA-DETRAC dataset but less than on the MOTChallenge dataset. The performance gain on the *camera blackout* experiments are particularly strong when only considering cyclists.

In the *prediction* experiments (see Appendix D), MOHART consistently outperforms HART. On both datasets, the model outperforms a baseline which uses momentum to linearly extrapolate the bounding boxes from the first two frames. This shows that even from just two frames, the model learns to capture motion models which are more complex than what could be observed from just the bounding boxes (i.e. momentum), suggesting that it uses visual information (HART & MOHART) as well as relational reasoning (MOHART).

## 6 CONCLUSION

With MOHART, we introduce an end-to-end multi-object tracker that is capable of capturing complex interactions and leveraging these for precise predictions as experiments both on toy and real world data show. However, the experiments also show that the benefit of relational reasoning strongly depends on the nature of the data. The toy experiments showed that in an entirely deterministic world relational reasoning was much less important than in a stochastic environment. Amongst the real-world dataset, the highest performance gains from relational reasoning were achieved on the MOTChallenge dataset, which features crowded scenes, ego-motion and occlusions.

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

## A  EXPERIMENTAL DETAILS

The MOTChallenge and the UA-DETRAC dataset discussed in this section are intended to be used as a benchmark suite for multi-object-tracking in a tracking-by-detection paradigm. Therefore, ground truth bounding boxes are only available for the training datasets. The user is encouraged to upload their model which performs tracking in a data association paradigm leveraging the provided bounding box proposals from an external object detector. As we are interested in a different analysis (IoU given inital bounding boxes), we divide the training data further into training and test sequences. To make up for the smaller training data, we extend the MOTChallenge 2017 dataset with three sequences from the 2015 dataset (ETH-Sunnyday, PETS09-S2L1, ETH-Bahnhof). We use the first 70% of the frames of each of the ten sequences for training and the rest for testing. Sequences with high frame rates (30Hz) are sub-sampled with a stride of two. For the UA-DETRAC dataset, we split the 60 available sequences into 44 training sequences and 16 test sequences. For the considerably larger Stanford Drone dataset we took three videos of the scene *deathCircle* for training and the remaining two videos from the same scene for testing. The videos of the drone dataset were also sub-sampled with a stride of two to increase scene dynamics.

## B  ARCHITECTURE DETAILS

The architecture details were chosen to optimise HART performance on the MOTChallenge dataset. They deviate from the original HART implementation (Kosiorek et al. (2017)) as follows: A presence variable predicts whether an object is in the scene and successfully tracked. This is trained with a binary cross entropy loss. The maximum number of objects to be tracked simultaneously was set to 5 for the UA-DETRAC and MOTChallenge dataset. For the more crowded Stanford drone dataset, this number was set to 10. The feature extractor is a three layer convolutional network with a kernel size of 5, a stride of 2 in the first and last layer, 32 channels in the first two layers, 64 channels in the last layer, ELU activations, and skip connections. This converts the initial $32 \times 32 \times 3$ glimpse into a $7 \times 7 \times 64$ feature representation. This is followed by a fully connected layer with a 128 dimensional output and an elu activation. The spatial attention parameters are linearly projected onto 128 dimensions and added to this feature representation serving as a positional encoding. The LSTM has a hidden state size of 128. The self-attention unit in MOHART comprises linear projects the inputs to dimensionality 128 for each keys, queries and values. For the real-world experiments, in addition to the extracted features from the glimpse, the hidden states from the previous LSTM state are also fed as an input by concatinating them with the features. In all cases, the output of the attention module is concatenated to the input features of the respective object.

As an optimizer, we used RMSProp with momentum set to $0.9$ and learning rate $5*10^{-6}$. For the MOTChallenge dataset and the UA-DETRAC dataset, the models were trained for 100,000 iterations of batch size 10 and the reported IoU is exponentially smoothed over iterations to achieve lower variance. For the Stanford Drone dataset, the batch size was increased to 32, reducing time to convergence and hence model training to 50,000 iterations.

## C  DETERMINISTIC TOY DOMAIN

In our first experiment in the toy domain (Figure 6), four circles each exert repulsive forces on each other, where the force scales with $1/r$, $r$ being their distance. HART is applied four times in parallel and is trained to predict the location of each circle three time steps into the future. The different forces from different objects lead to a non-trivial force field at each time step. Predicting the future location just using the previous motion of one object (Figure 6 shows that each spatial attention box covers only the current object) accurately is therefore challenging. Surprisingly, the single object tracker solves this task with an average of $95\%$ IoU over sequences of 15 time steps. This shows the efficacy of end-to-end tracking to capture complex motion

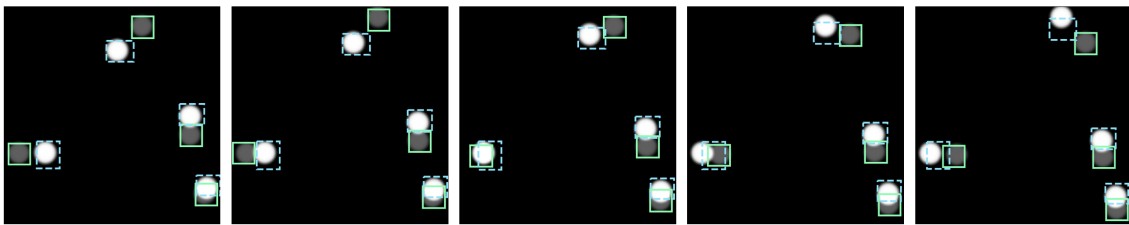

FIGURE 6: HART single object tracking applied four times in parallel and trained to predict the location of each circle three time steps into the future. Dashed lines indicate spatial attention, solid lines are predicted bounding boxes, faded circles show ground truth location at $T + 3$. Each circle exerts repulsive forces on each other, where the force scales with $1/r$, $r$ being their distance.

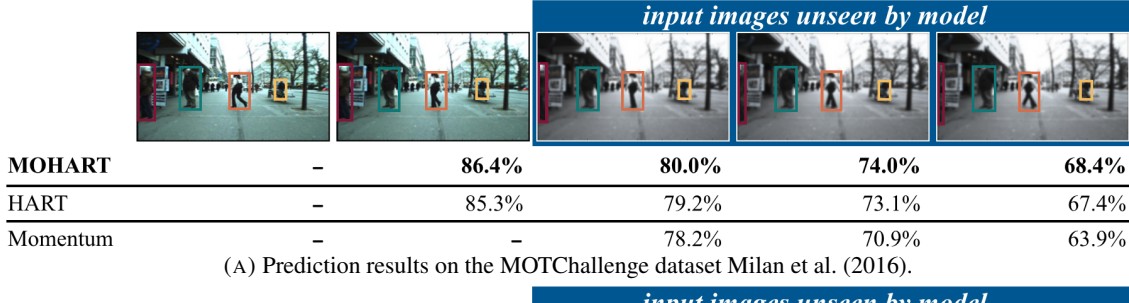

| | | input images unseen by model | | |
|---|---|---|---|---|
| **MOHART** | – | **86.4%** | **80.0%** | **74.0%** | **68.4%** |
| HART | – | 85.3% | 79.2% | 73.1% | 67.4% |
| Momentum | – | – | 78.2% | 70.9% | 63.9% |

(A) Prediction results on the MOTChallenge dataset Milan et al. (2016).

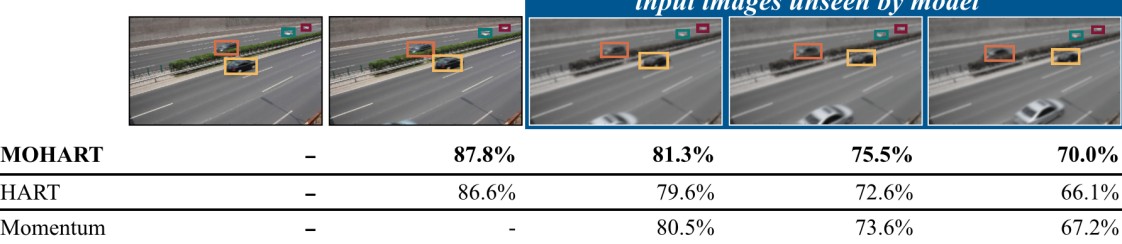

| | | input images unseen by model | | |
|---|---|---|---|---|
| **MOHART** | – | **87.8%** | **81.3%** | **75.5%** | **70.0%** |
| HART | – | 86.6% | 79.6% | 72.6% | 66.1% |
| Momentum | – | - | 80.5% | 73.6% | 67.2% |

(B) Prediction results on the UA-DETRAC dataset (crowded scenes only) Wen et al. (2015).

FIGURE 7: Peeking into the future. Only the first two frames are shown to the tracking algorithm followed by three black frames. MOHART learns to fall back on its internal motion model when no observation (i.e. only a black frame) is available. The reported IoU scores show the performance for the respective frames 0, 1, 2, and 3 time steps into the future.

patterns and use them to predict future locations. This, of course, could also be used to generate robust bounding boxes for a tracking task.

## D   PREDICTION EXPERIMENTS

In the results from the *prediction* experiments (see Figure 7) MOHART consistently outperforms HART. On both datasets, the model outperforms a baseline which uses momentum to linearly extrapolate the bounding boxes from the first two frames. This shows that even from just two frames, the model learns to capture motion models which are more complex than what could be observed from just the bounding boxes (i.e. momentum), suggesting that it uses visual information (HART & MOHART) as well as relational reasoning (MOHART). The strong performance gain of MOHART compared to HART on the UA-DETRAC dataset,

despite the small differences for tracking on this dataset, can be explained as follows: this dataset features little interactions but strong correlations in motion. Hence when only having access to the first two frames, MOHART profits from estimating the velocities of multiple cars simultaneously.

## E    QUALITATIVE TRACKING RESULTS

**HART (Single-Object Tracking)**

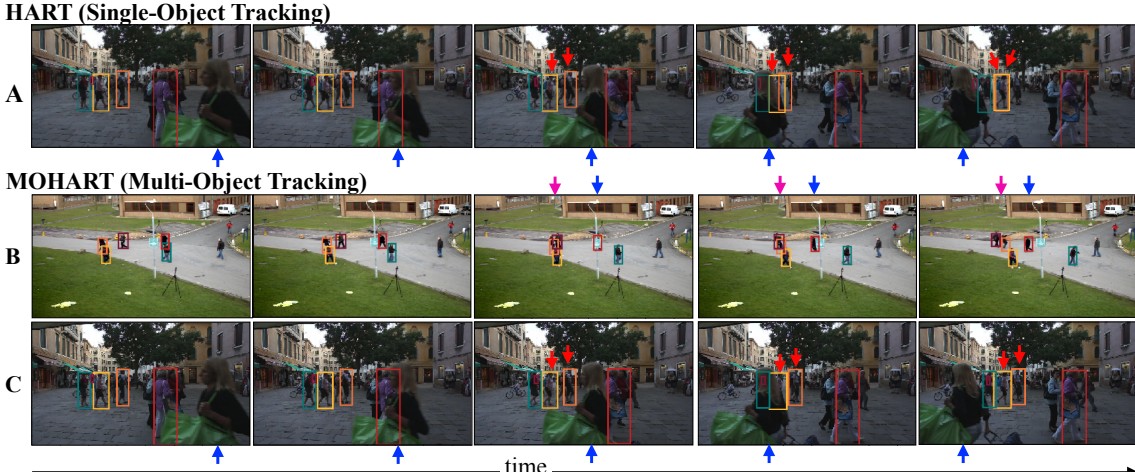

**MOHART (Multi-Object Tracking)**

FIGURE 8: Tracking examples of both HART and MOHART. Coloured boxes are bounding boxes predicted by the model, arrows point at challenging aspects of the scenes. (A) & (C): Each person being tracked is temporarily occluded by a woman walking across the scene (blue arrows). MOHART, which includes a relational reasoning module, handles this more robustly (compare red arrows).

In Section 5, we tested MOHART on three different real world data sets and in a number of different setups. Figure 8 shows qualitative results both for HART and MOHART on the MOTChallenge dataset.

Furthermore, we conducted a set of camera blackout experiments to test MOHART's capability of dealing with faulty sensor inputs. While traditional pipeline methods require careful consideration of different types of corner cases to properly handle erroneous sensor inputs, MOHART is able to capture these automatically, especially when confronted with similar issues in the training scenarios. To simulate this, we replace subsequences of the images with black frames. Figure 9 and Figure 5 show two such examples from the test data together with the model's prediction. MOHART learns not to update its internal model when confronted with black frames and instead uses the LSTM to propagate the bounding boxes. When proper sensor input is available again, the model uses this to make a rapid adjustment to its predicted location and 'snap' back onto the object. This works remarkably well in both the presence of occlusion (Figure 9) and ego-motion (Figure 5). Tables 1 to 3 show that the benefit of relational reasoning is particularly high in these scenarios specifically. These experiments can also be seen as a proof of concept of MOHART's capabilities of predicting future trajectories—and how this profits from relational reasoning.

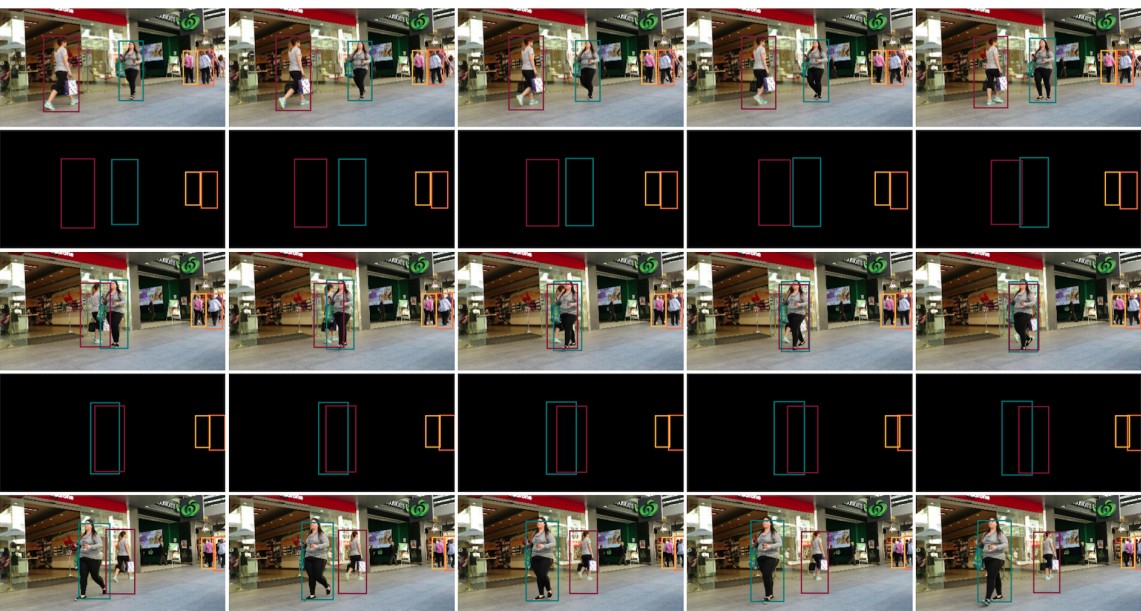

FIGURE 9: Camera blackout experiment on a pedestrian street scene from the MOTChallenge dataset without ego-motion. Subsequent frames are displayed going from top left to bottom right. Shown are the inputs to the model (some of them being black frames, i.e. arrays of zeroes) and bounding boxes predicted by MOHART (coloured boxes). This scene is particularly challenging as occlusion and missing sensor input coincide (fourth row).

