# OpenReview forum: "Improving End-to-End Object Tracking Using Relational Reasoning"
_ICLR.cc/2020/Conference — Reject_

### Official Review · AnonReviewer1 · 2019-10-22
**Official Blind Review #1**

**Rating:** 6

**Review:**

I'm somewhat out of area for this review: I study relational models, but have little experience with computer vision in general and object tracking in particular.

--------------------

This paper presents an extension to HART [Kosiorek 2017] to track multiple objects. It builds on a simple baseline: run multiple HART models in parallel, each tracking one of the objects. They extend this by adding a relational reasoning module to allow interaction between the parallel models. The relational reasoning module uses Lee et al. (2019)'s self-attention block (similar to Vaswani et al. 2017). They find that the simple baseline is surprisingly effective, but that MOHART (their model) improves performance in environments with stochastic interactions and crowded settings which tend to be more noisy.

The novelty is somewhat limited: they're plugging together two existing approaches (HART and SAB) to allow for interaction, but I am recommending acceptance because I found the experiments surprising (as a negative result) - they show that independent HART models are a strong baseline for tracking in settings that involve interactions, but that are nevertheless solvable without knowledge about the state of other objects. For the synthetic experiments, I would have expected the fact there are repulsive forces between the objects would have been sufficient for the relational module to be helpful. The paper also provides further evidence of the usefulness of non-local networks for these sorts of problems.

**Experience Assessment:**

I do not know much about this area.

**Review Assessment: Checking Correctness Of Derivations And Theory:**

N/A

**Review Assessment: Checking Correctness Of Experiments:**

I assessed the sensibility of the experiments.

**Review Assessment: Thoroughness In Paper Reading:**

I read the paper at least twice and used my best judgement in assessing the paper.

---

> ### Author Response · Authors · 2019-11-12
> **Response to R1**
>
> We are delighted that the reviewer finds that the experiments and results are a worthwhile contribution to the ICLR community.
>
> As regards the structural novelty, beyond the addition of the relational reasoning module, another structural contribution is the ability to track multiple objects in parallel. This is thanks to a new presence variable, which indicates whether an object is (still) in the scene. Beyond being useful as an output of the model, the ground-truth presence is also used for masking the training loss. We believe that the simplicity of using two well-understood components (HART end-to-end tracking and Transformer-based attention) with a clear objective (tracking with relational reasoning) is an advantage. The modularity of the relational reasoning component has the further benefit of being easily applicable to future generations of end-to-end tracking.
>
> The reason that fixed repulsive forces are well-modeled by HART is its LSTM-based state estimation: since the forces change deterministically, a single LSTM cell can predict their evolution. We believe that performance gains from relational reasoning would be higher if HART used a weaker state-estimation mechanism.

---

> > ### Comment · AnonReviewer1 · 2019-11-12
> > **Response to authors**
> >
> > Agreed re structural novelty. Also I actually meant to write "The [architectural] novelty is"... to be clear, I don't think we as a community should be optimizing for architectural novelty, which is part of the reason I voted to accept this paper. Good empirical evaluations of novel combinations of existing ideas are useful.

---

> > > ### Author Response · Authors · 2019-11-15
> > > **Response to R1**
> > >
> > > Couldn't agree more! :)

---

### Official Review · AnonReviewer3 · 2019-10-22
**Official Blind Review #3**

**Rating:** 3

**Review:**

The paper presents a class-agnostic method for tracking multiple moving objects (MOHART) that extends an existing single-object tracking method (Hierarchical Attentive Recurrent Tracking, HART). Similarly to HART, MOHART utilizes an attention mechanism and LSTM units. The extension form HART to MOHART is done in two main steps: HART is applied to multiple objects in parallel, with a presence variable attached to each, and a permutation-invariant network that learns the interactions between the objects.

The method is tested in two groups of experiments: synthetic data of moving circles, and on several real datasets. The results on the moving circles are enlightening. The paper shows that although MOHART outperforms HART in all circumstances, if the forces between the circles change randomly every number of steps, MOHART outperforms HART by a very large margin. This is because the forces between the circles are determined by their color, which is observable by the network, and MOHART, which capable of learning their interaction, learns the pattern of changing forces. HART, being an independent-particle model, cannot learn the interactions.

The results on the real datasets demonstrate superiority over HART, and in a series of ablation experiments, the Authors are able to show that MOHART's ability to utilize interactions between objects is responsible for the improvement. For example, the improvement over HART is greater when egomotion is significant, because egomotion creates correlations between the movement of objects in the camera frame, which MOHART can utilize.

I found the paper enlightening, based on a neat idea. The experimental results are extensively analyzed and the Author's insights about their algorithm are substantiated by the experiments.

However the difficulty I had with the paper is that the results are only compared to HART. HART was published in 2017, and the field of scene understanding and tracking had been addressed in multiple papers since then. It is one of the more competitive and application-driven fields in computer vision. Comparing MOHART only to its sister-method from over two years ago significantly limits the usefulness of paper IMHO. The expectation IMHO is not that MOHART outperform each and every method out there, but that the reader know where MOHART stands compared to other methods (e. g. tracking by detection).

I will refrain from pointing to specific papers to compare to, because I believe the Authors should chose the settings, datasets, criteria, and metrics that are the most convenient for them for comparison. The papers surveyed in Section 2, especially under "Tracking-by-Detection", have many followups. Recent papers from those that claim state of the art could good candidates for comparison.

In view of the above, I am inclined at this stage to reject the paper.

**Experience Assessment:**

I have read many papers in this area.

**Review Assessment: Checking Correctness Of Derivations And Theory:**

N/A

**Review Assessment: Checking Correctness Of Experiments:**

I assessed the sensibility of the experiments.

**Review Assessment: Thoroughness In Paper Reading:**

I read the paper at least twice and used my best judgement in assessing the paper.

---

> ### Author Response · Authors · 2019-11-12
> **Response to R3**
>
> We are pleased that the author finds the idea neat and the findings of the paper enlightening.
>
> As regards the comparison of MOHART to other SOTA approaches: while it is true that the field has evolved since the publication of HART, we are not aware of recent end-to-end tracking approaches for real-world data, and especially we were unable to find ones that track multiple objects at a time.
>
> We note that it is not straight-forward to quantitatively compare the performance of (MO)HART and non-end2end methods such as tracking-by-detection algorithms. MOHART is evaluated with ground truth bounding boxes provided in the first video frame with all objects present. This evaluation paradigm is used, e.g., in the Visual Object Tracking Challenge (Kristan et al. 2016) where performance is reported as intersection over union. This dataset, however, contains only one annotated object per sequence and is unsuitable for multi-object tracking evaluation. We are not aware of any dataset or benchmark for multi-object tracking that would use a similar evaluation protocol. The MOTChallenge dataset, on the other hand, which contains many annotated objects per frame, is set up as a data association problem. Instead of ground truth bounding boxes in the first frame, the dataset provides (imperfect) detections for each individual frame. The goal of tracking-by-detection algorithms is to match the detections of the same object across frames. Algorithms typically applied to this dataset would have an unfair advantage over MOHART due to the access to external detections, and the results of such a comparison could be misleading for a potential reader.
>
> Having said that, we do agree that having a broader comparison with other approaches would be helpful for the readers to better position MOHART within the existing literature. We are currently looking into possible datasets and baselines to provide further comparison.

---

### Official Review · AnonReviewer2 · 2019-10-24
**Official Blind Review #2**

**Rating:** 3

**Review:**

This paper deals with the problem of multiple object tracking and trajectory prediction in multiple frames of videos. The main focus is adding a relation-reasoning building block to the original HART framework. With multiple objects, the key is to be able to learn the permutation invariant representation during potential changing and dynamic object trajectories. The paper also uses toy examples to show that the proposed block of relation reasoning is not necessarily beneficial when the object trajectory is less random and more static. Finally, experiments on real data demonstrate that the proposed method that accounts for relation reasoning is helpful by a limited magnitude.

The main contribution of this paper is the novel relation reasoning block. However, there are three key concerns of mine:

1. It is not clear when the new architecture would be helpful. I am really happy to see a more careful study of the success and failure cases of the method. I also appreciate the honesty that the current model is not competitive with the ones that have an accurate bounding box as input. But I think a more detailed study can be conducted, especially in the toy example case.  Maybe it is possible to design different levels of randomness in the trajectory and further figure out when is the reasoning block helpful.

2. Even the real data experiments do not have very impressive results, from my biased observation. (I am not in the field, so maybe I am wrong here.)

3. The end-to-end approach is also another perspective where the authors try to differentiate their methods from others. I think it is potentially an interesting problem that why end-to-end is something you would prefer (which is definitely a harder problem), and when.


**Experience Assessment:**

I do not know much about this area.

**Review Assessment: Checking Correctness Of Derivations And Theory:**

N/A

**Review Assessment: Checking Correctness Of Experiments:**

I assessed the sensibility of the experiments.

**Review Assessment: Thoroughness In Paper Reading:**

I made a quick assessment of this paper.

---

> ### Author Response · Authors · 2019-11-12
> **Response to R2**
>
> 1. "It is not clear when the new architecture would be helpful. […] I think a more detailed study can be conducted, especially in the toy example case.  Maybe it is possible to design different levels of randomness in the trajectory and further figure out when is the reasoning block helpful."
>
> Could you please elaborate on what you mean by different levels of randomness? What you are describing sounds to us very similar to figure 4b: we vary the level of randomness (i.e. how often the electron/proton identity is randomly re-assigned). The experiments show that the higher the stochasticity, the more important the relational reasoning. In a fully deterministic version of this environment, where the forcefield can be inferred from the trajectory of one particle, relational reasoning is not necessary. We believe that this insight matches well with the results from the real-world experiments: e.g., camera ego-motion could be seen as a stochastic external factor, leading to bigger performance gains from relational reasoning.
>
> If you had a different experiment in mind, we would be very curious to hear about your idea!
>
> 2. "Even the real data experiments do not have very impressive results, from my biased observation. (I am not in the field, so maybe I am wrong here.)"
>
> We personally do find of the real-world results ‘impressive’ (being fully aware of the subjectiveness of this statement), especially figures 5 and 9 where sensor input is only partially available. While no new sensor output is available (feeding just black frames to the model), the algorithm learns to fall back onto its internal motion model, predicting how the pedestrians walk. Once new sensor data is available, the model ‘snaps back’ onto the objects. This is learned fully end-to-end while not applying any algorithmic changes to MOHART compared to the vanilla tracking setup.
>
> 3. "The end-to-end approach is also another perspective where the authors try to differentiate their methods from others. I think it is potentially an interesting problem that why end-to-end is something you would prefer (which is definitely a harder problem), and when."
>
> The strength of tracking-by-detection approaches lies in their rigidity because they can be carefully designed to fit the particular challenges of the problem. Particularly with known object classes and a plethora of training data the external object detector can provide a very strong starting point. End-to-end tracking, on the other hand, has the benefit of being very flexible: It can be applied to track any object class, even one that was never seen during training time. The algorithm also has the power of learning about a wide range of phenomena such as ego-motion and faulty sensor inputs (both of which particularly benefit from relational reasoning, as our experiments show).

---

### Author Response · Authors · 2020-05-13
**Updated Paper**

We thank the area chair and the reviewers for their time, effort, and their constructive feedback. We used the feedback to update our paper in the following way: We specified our contributions and specifically pointed out the advantages of end-to-end tracking with an internal motion model and relational reasoning over classical methods. We also added references to Frossard & Urtasun as well as the most recent state-of-the-art visual object trackers.

Most importantly, we added a quantitative comparison to state-of-the-art visual object trackers.

The new version can be found on arxive:

https://arxiv.org/pdf/1907.12887.pdf

---

### Decision · Program_Chairs · 2019-12-19

**Decision:**

Reject

**Comment:**

The authors propose an end-to-end object tracker by exploiting the attention mechanism. Two reviewers recommend rejection, while the last reviewer is more positive. The concerns brought up are novelty (last reviewer), and experiments (second reviewer). Furthermore, the authors seem to overclaim their contribution. There indeed are end-to-end multi-object trackers, see Frossard & Urtasun's work for example. This work needs to be cited, and possibly a comparison is needed. Since the paper did not receive favourable reviews and there are additional citations missing, this paper cannot be accepted in current form. The authors are encouraged to strengthen their work and resubmit to a future venue.